

# Fire weather effects on flammability of indigenous and invasive alien plants in coastal fynbos and thicket shrublands (Cape Floristic Region)

Samukelisiwe T. Msweli[1,*], Alastair J. Potts[2], Herve Fritz[3,4] and Tineke Kraaij[1,*]

[1] School of Natural Resource Management, Nelson Mandela University, George, Western Cape, South Africa
[2] Botany Department, Nelson Mandela University, Port Elizabeth, Eastern Cape, South Africa
[3] REHABS International Research Laboratory, CNRS-Université de Lyon1-Nelson Mandela University, George, South Africa
[4] Sustainability Research Unit, Nelson Mandela University, George, Western Cape, South Africa
* These authors contributed equally to this work.

Corresponding author
Tineke Kraaij,
tineke.kraaij@mandela.ac.za

## ABSTRACT

**Background:** Globally, and in the Cape Floristic Region of South Africa, extreme fires have become more common in recent years. Such fires pose societal and ecological threats and have inter alia been attributed to climate change and modification of fuels due to alien plant invasions. Understanding the flammability of different types of indigenous and invasive alien vegetation is essential to develop fire risk prevention and mitigation strategies. We assessed the flammability of 30 species of indigenous and invasive alien plants commonly occurring in coastal fynbos and thicket shrublands in relation to varying fire weather conditions.
**Methods:** Fresh plant shoots were sampled and burnt experimentally across diverse fire weather conditions to measure flammability in relation to fire weather conditions, live fuel moisture, fuel load and vegetation grouping (fynbos, thicket and invasive alien plants). Flammability measures considered were: burn intensity, completeness of burn, time-to-ignition, and the likelihood of spontaneous ignition. We also investigated whether the drying of plant shoots (simulating drought conditions) differentially affected the flammability of vegetation groups.
**Results:** Fire weather conditions enhanced all measures of flammability, whereas live fuel moisture reduced burn intensity and completeness of burn. Live fuel moisture was not significantly correlated with fire weather, suggesting that the mechanism through which fire weather enhances flammability is not live fuel moisture. It furthermore implies that the importance of live fuel moisture for flammability of evergreen shrublands rests on inter-specific and inter-vegetation type differences in fuel moisture, rather than short-term intra-specific fluctuation in live fuel moisture in response to weather conditions. Fuel load significantly increased burn intensity, while reducing ignitability. Although fire weather, live fuel moisture, and fuel load had significant effects on flammability measures, vegetation and species differences accounted for most of the variation. Flammability was generally highest in invasive alien plants, intermediate in fynbos, and lowest in thicket. Fynbos ignited rapidly and burnt completely, whereas thicket was slow to ignite and burnt

incompletely. Invasive alien plants were slow to ignite, but burnt with the highest intensity, potentially due to volatile organic composition. The drying of samples resulted in increases in all measures of flammability that were comparable among vegetation groups. Flammability, and by implication fire risk, should thus not increase disproportionately in one vegetation group compared to another under drought conditions—unless the production of dead fuels is disproportionate among vegetation groups. Thus, we suggest that the dead:live fuel ratio is a potentially useful indicator of flammability of evergreen shrublands and that proxies for this ratio need to be investigated for incorporation into fire danger indices.

## INTRODUCTION

Flammability is the ability of vegetation (fuel) to burn (*Fernandes & Cruz, 2012*; *Gill & Zylstra, 2005*) and is a measure of fire behavior (fire intensity/severity) used in vegetation fire risk mitigation studies (*Keeley, 2009*). Vegetation flammability may result from climatic and weather effects (*Bond & Midgley, 1995*; *Mutch, 1970*; *Snyder, 1984*). For example, in arid areas, dryness limits fuel accumulation and fires follow episodic rain, whereas in temperate areas, fuel loads are not limiting but fires follow the drying of those fuels (*Bradstock, 2010*; *Pausas & Bradstock, 2007*), however dry conditions may also result in an increase in fire risk caused by the availability of dried fuels (*Piñol, Terradas & Llored, 1998*). Fire-prone vegetation groups may furthermore have evolved traits that enhance their flammability and improve vegetation fitness in fire-dependent communities (*Bond & Midgley, 1995*). Correspondingly, species with high flammability traits may burn intensely, such that itself and the neighbour die, thereby facilitating recruitment—the "kill thy neighbour" hypothesis (*Bond & Midgley, 1995*). Flammability traits may thus provide resilience associated with fire tolerance (*Bond & Midgley, 1995*; *Calitz, Potts & Cowling, 2015*). Fire is accordingly one of the main determining factors of the ecology and distribution of ecosystems of the world, and is important for maintaining plant diversity (*Bond, 1997*; *Bond, Midgley & Woodward, 2003*; *Bond & Keeley, 2005*).

Flammability is affected by weather conditions (*Bond, 1997*; *Keeley & Syphard, 2017*). Fire danger indices—based on ambient temperature, relative humidity, wind speed, and rainfall—are commonly used to rate the fire-proneness of weather conditions (*Dowdy et al., 2009*; *Noble, Gill & Bary, 1980*; *Sirca et al., 2018*). Flammability is also influenced by fuel properties such as the amount of flammable plant material (fuel load), packing ratio and chemical composition (*Brooks et al., 2004*; *Burger & Bond, 2015*; *Curran et al., 2017*). For instance, greater fuel loads or volatile substances can increase fire intensity (*Baeza et al., 2002*; *Saura-Mas et al., 2010*).

Globally, extreme fires have become more common in recent years. Examples include the shrublands of California, Australia, Europe (*Montenegro et al., 2004*; *San-Miguel-Ayanz, Moreno & Camia, 2013*), and more recently, South Africa (*Kraaij et al., 2018*).

These fires have been accredited to the combinations of climate change (in the form of weather conditions more conducive to fire and extended droughts), increased ignitions, expanded wildland-urban interface areas linked to increasing human populations, and changes in fuels that are often human-induced (*Archibald et al., 2008*; *Montenegro et al., 2004*; *Syphard et al., 2017*; *Turco et al., 2017*; *Van Wilgen, 1984*). Fuels accumulate far beyond normal levels when humans suppress fires to safeguard assets, and due to invasion by invasive alien plants (hereafter IAPs) (*Kraaij et al., 2018*; *Radeloff et al., 2005*; *Scott, Versfeld & Lesch, 1998*). The IAPs may affect flammability by altering the fuel structure, fuel distribution (horizontal or vertical fuel continuity), live fuel moisture, chemical contents and fuel load (*Brooks et al., 2004*; *Davies & Nafus, 2013*; *Richardson & Van Wilgen, 2004*). Extreme fires are also known to occur in shrublands after severe droughts due to the increase of dead (~dry) to live fuel ratios (*Keeley et al., 2012*; *Keeley & Syphard, 2017*; *Kraaij et al., 2018*).

Along the southern Cape coast of South Africa, fynbos and thicket shrublands occur interspersed (Fig. S1) despite displaying different fire dynamics and fuel structural traits (*Campbell et al., 1981*; *Moll et al., 1984*). Fynbos ecosystems commonly support fires that consume surface and canopy fuels and comprise species that readily burn to open recruitment opportunities (gaps) post-fire (*Buhk, Meyn & Jentsch, 2007*; *Deacon, Jury & Ellis, 1992*). However, thicket mostly does not exhibit high flammability traits (*Calitz, Potts & Cowling, 2015*), and recruitment from seed largely occurs in inter-fire periods (*Cowling et al., 1997*; *Pierce & Cowling, 1984*). In 2017, extreme fires occurred in this region around the town of Knysna which burnt indigenous fynbos and thicket vegetation and further caused extensive damage to commercial plantations and residential properties (*Fares et al., 2017*; *Kraaij et al., 2018*). The extreme nature of these fires has been attributed to extensive IAP fuels, an expansive wildland-urban interface area, an unprecedented regional drought preceding the fires which likely greatly increased litter fuels in thicket, fynbos and stands of IAPs, and very high fire danger weather conditions at the time of the fires (*Kraaij et al., 2018*; *Preston, 2017*). The 2017 Knysna fires called for improved understanding of potential differences in flammability among vegetation groups, including IAPs occurring in this region. An analysis of satellite image derived proxies for burn severity showed it to be higher, but completeness of burn lower, in IAPs than in indigenous fynbos and thicket vegetation (*Kraaij et al., 2018*). However, the findings have not been verified with field observations (*Kraaij et al., 2018*). Other studies have experimentally compared the flammability of species from several biomes (both fire-prone and fire-resistant) (*Burger & Bond, 2015*; *Calitz, Potts & Cowling, 2015*); however, no study has compared the flammability of indigenous vegetation with that of IAPs, nor under varying fire weather conditions.

The primary aim of our study was to compare the flammability of live plant material amongst three vegetation groups—IAPs, fynbos, and thicket—and under varying fire weather conditions. Flammability measures considered were burn intensity, completeness of burn, and ignitability (time-to-ignition and likelihood of spontaneous ignition), while fuel traits considered were live fuel moisture and fuel load. A secondary aim was to assess the flammability of partially dried plant material as a crude proxy for drought

effects, to ascertain whether drying of fuels (~drought) would differentially affect the flammability of the vegetation groups of interest. Study results will inform fire risk management in the southern Cape landscapes and elsewhere with similar fuel traits and characteristics.

## MATERIALS AND METHODS

### Study area

This study was conducted along the southern Cape coast of South Africa within the Cape Floristic Region close to the city of George (33.964°S, 22.534°E). The climate is moderated by the maritime influence with average minimum and maximum temperatures ranging from 7 °C to 19 °C in June and 15 °C to 26 °C in January, and annual average rainfall of approximately 800 mm throughout the year (*Bond, 1981*). The area experiences weather conditions suitable for fires at any time of the year and fires are often associated with hot, dry katabatic ("berg") winds (*Kraaij, Cowling & Van Wilgen, 2013*; *Van Wilgen, 1984*).

The vegetation of the study area is classified as Southern Cape Dune Fynbos (*Mucina & Rutherford, 2006*; *Pierce & Cowling, 1984*), which consists of medium-dense sclerophyllous fynbos (~fine-leaved) shrublands up to 2 m in height, interspersed with dense clumps of subtropical mesophyllous thicket shrubs or trees up to 4 m in height (Fig. S1) (*Campbell et al., 1981*; *Kraaij, Cowling & Van Wilgen, 2011*; *Pierce & Cowling, 1984*). Both fynbos and thicket are evergreen. Fynbos shrublands are fire-prone and flammable while smaller areas of thicket vegetation seldom burn (*Geldenhuys, 1994*). The persistence of fynbos-thicket mosaics requires fire at appropriate intervals (15–25 years) since thicket becomes dominant in the prolonged absence of fire (*Kraaij & Van Wilgen, 2014*; *Strydom et al., 2020*). The area contains extensive invasions of alien shrubs and trees, commonly of the genera *Acacia, Eucalyptus* and *Pinus*, that co-occur with, and potentially replace, the native vegetation (Fig. S1) (*Baard & Kraaij, 2014*; *Van Wilgen et al., 2016*).

### Data collection

#### Live plant samples

We experimentally measured the flammability of plant shoots (i.e., plant stems) of species from three vegetation groups, namely IAPs, fynbos and thicket. Fine fuels such as plant shoots (hereafter samples) are the primary carriers or vectors of fire spread (*Murray, Hardstaff & Phillips, 2013*). Our experiments were thus focused on plant shoots, with one stem constituting one sample. Sampling was done over 21 occasions (February–November 2018) that were specifically selected to represent varying fire weather conditions. On each occasion, we collected two live plant samples of 30 species across three vegetation groups (10 species per vegetation group; details in Table S1) common in the study area. One sample was used for flammability experiments, while the other was used for live fuel moisture measurements. For each species, samples of approximately 70 cm in length that were representative of the fuel structure characteristic of the species were sourced. On each sampling occasion, samples from all 30 species were collected

and burnt to ensure that flammability was measured under comparable conditions. Sample collection either started at 09:00 and subsequent burning at 12:00 or at 11:00 and 14:00 (respectively) to incorporate additional variation in fire weather conditions. Samples were kept in closed plastic containers after collection prior to burning, and burning was completed within four hours of sample collection to minimise moisture loss. The order in which samples of different species were burnt was also randomised among the different burning occasions to not consistently expose particular species to longer periods of moisture loss prior to burning. For each occasion, the Canadian fire weather index was computed based on the temperature, relative humidity, rainfall (over the past 24 h), and wind speed (*Van Wagner & Forest, 1987*) at the time that burning commenced. This index integrates drought and other atmospheric effects that are relevant to fire behavior and fuel moisture, and it was shown to be the best performing fire danger index in Mediterranean ecosystems (*Sirca et al., 2018*). The input weather measures were obtained from a weather station located on the George Campus of Nelson Mandela University ("Saasveld NMMU CW373" on the Vital Weather online platform: www.vitalweather.co.za) where the experimental burning was conducted.

Samples used for flammability were burnt outdoors using an approach similar to that of *Calitz, Potts & Cowling (2015)* and *Curran et al. (2017)*. Plant flammability was measured using the method and equipment described by *Jaureguiberry, Bertone & Díaz (2011)*, the apparatus comprises a metal barrel (85 cm × 60 cm) that is horizontally orientated with the top removable half that is used for wind protection (*Baeza et al., 2002*). The metal barrel is connected to a grill thermometer, removable gas cylinder and a blowtorch (*Curran et al., 2017*; *Jaureguiberry, Bertone & Díaz, 2011*). Each sample was placed on the barrel cavity grill to pre-heat at 230 °C for 2 min to imitate the heating and drying effect of an approaching fire. If the samples had not spontaneously ignited within two minutes, it was ignited at the top of the shoot by exposing it to the blow torch for a period of five seconds (*Calitz, Potts & Cowling, 2015*). Advantages of using this apparatus are that it preserves the architectural arrangement of plant material (*Jaureguiberry, Bertone & Díaz, 2011*). It further enables a more realistic comparison of relative canopy flammability among species than methods that use only smaller plant components (i.e., twigs or leaves) (*Burger & Bond, 2015*; *Jaureguiberry, Bertone & Díaz, 2011*).

Four aspects associated with species-level flammability were measured and recorded (largely after *Calitz, Potts & Cowling, 2015* and *Jaureguiberry, Bertone & Díaz, 2011*). Firstly, burn intensity taken as the maximum temperature (cf. *Keeley, 2009*) reached by a sample while burning, measured using an infrared thermometer (Major Tech 695; maximum recordable temperature: 800 °C) after *Jaureguiberry, Bertone & Díaz (2011)*. Secondly, the completeness of burn, calculated as the proportion of the pre-burn wet mass of the samples that was consumed by the fire (mass was measured using an electronic scale). Thirdly, time-to-ignition, measured as the time elapsed between placement of the samples on the grill and spontaneous ignition (appearance of the first flame); samples that required to be ignited with the blow torch were therefore excluded from this measure in the

dataset. For every sample, we recorded whether it spontaneously ignited within the two minutes (pre-heating duration was consistent as there were many samples) of pre-heating or not, this binomial response comprising the fourth measure termed "spontaneous ignition".

Live fuel moisture was calculated on a sample shoot similar in dimensions to that of flammability measurements. The fresh material was stored in sealed containers (of known mass) until these were weighed (within less than 3 h of collection) to obtain wet fuel mass. Samples were then oven-dried at 80 °C for 48 h and weighed again to obtain dry fuel mass (*Ruffault et al., 2018*; *Teie, 2009*; *Yebra et al., 2019*). The live fuel moisture was calculated as the percentage of wet mass comprised of water. Although sample size (shoot length) was standardized, samples nevertheless presented different fuel loads which are directly related to burn intensity (*Byram, 1959*). Thus, dry plant mass was used to represent the variable fuel load. We estimated the dry plant mass of each sample from its pre-burn wet mass and the percentage water content that was calculated for its analogous dried sample.

### Dried plant samples

To investigate whether simulated drought conditions differentially affected the flammability of the vegetation groups, additional samples (similar to that collected for the flammability experiment's live samples described above) were collected and left to dry under ambient conditions, out of direct sunlight, for a minimum of two weeks but not until leaf loss occurred. Sampling was conducted over five occasions (during February–March 2019) of high fire weather conditions. The drying duration was standardized for all species within each of the sampling occasions to avoid the loss of leaves since certain plants would drop leaves due to drought stress (*Clarke & McCaig, 1982*). Flammability experiments and pre-burn estimations of live fuel moisture were undertaken on these dried samples as described above for live (undried) samples.

## Data analysis

### Live plant samples

We assessed flammability (of live samples) in terms of four response variables (burn intensity, completeness of burn, time-to-ignition, and spontaneous ignition) respectively, in relation to the predictor variables (i) fire weather (continuous), (ii) live fuel moisture (continuous), (iii) fuel load (dry plant mass; continuous), (iv) vegetation groups (IAPs, fynbos, thicket; categorical) and (v) species (30 species; categorical) using generalized linear mixed-effects models (*Bates, 2010*; *O'Hara, 2009*) using the *lme4* package (*Bates, 2010*) in the open-source R software version 3.6.1 (*R Development Core Team, 2019*). Detailed species-level comparisons were not the primary focus of the study and species was therefore included as a random factor, whereas the other predictor variables were included as fixed factors. To test for potential collinearity between fire weather and live fuel moisture, we ran the Spearman-rank correlation test for each respective species. It showed that these variables were not significantly correlated (see Results) and could

both be retained in subsequent analyses. Burn intensity was log-transformed (to correct right-skewed distribution), completeness of burn arcsine-transformed (as it was expressed as proportions), time-to-ignition square root-transformed (to correct left-skewed distribution), and spontaneous ignition assessed using logistic regression (binomial family, logit link function) (formulae provided in Table S2). Subsequently, Type II Wald chi-square test (*Hastie & Pregibon, 1992*) was computed to determine the significance of fixed factors on the specific models. We incorporated the scale function to the generalized linear mixed-effects models and logistic regression model (using transformed data) to standardize variables of different scales and obtain the relative influence of fixed factors (*Becker, Chambers & Wilks, 2018*).

### Dried plant samples

We compared the flammability (in terms of burn intensity, completeness of burn, and time-to-ignition, respectively) of the dried samples with that of live samples of the same species that was measured on five occasions under comparable fire weather conditions. We calculated the change in flammability between live and dried samples by subtracting the flammability measure of each live sample from that of its dried counterpart. We then used this derived variable as response variable and employed Kruskal-Wallis to test whether the difference in flammability between live and dried samples varied among vegetation groups.

## RESULTS

### Live plant samples

Fire weather and live fuel moisture were not significantly correlated within any of the study species (Table S1). Increasing severity of fire weather significantly increased flammability through increasing burn intensity, increasing completeness of burn, increasing the likelihood of spontaneous ignition, and reducing time-to-ignition (Table 1; Fig. 1). Increasing live fuel moisture significantly decreased burn intensity, completeness of burn, and the likelihood of spontaneous ignition. Fuel load significantly increased burn intensity and time-to-ignition.

In considering vegetation groups, flammability was generally highest in IAPs, intermediate in fynbos, and lowest in thicket (Table 1; Fig. 1). IAPs burnt at significantly higher intensity than fynbos and thicket. IAPs and fynbos showed significantly higher ignitability (shorter time-to-ignition and a greater likelihood of spontaneous ignition) than thicket.

Amongst the different fixed factors, vegetation groups consistently had the largest influence (i.e., the largest scaled estimates; Table 1) on all flammability measures. Fire weather had the second largest influence on ignitability, while live fuel moisture had the second largest influence on burn intensity and completeness of burn.

The total variance in the flammability measures explained by the models was generally low (24–40%; conditional $R^2$ values, Table 1). The fixed factors combined explained less variation (8–22%; marginal $R^2$ values, Table 1) than species as random factor by itself
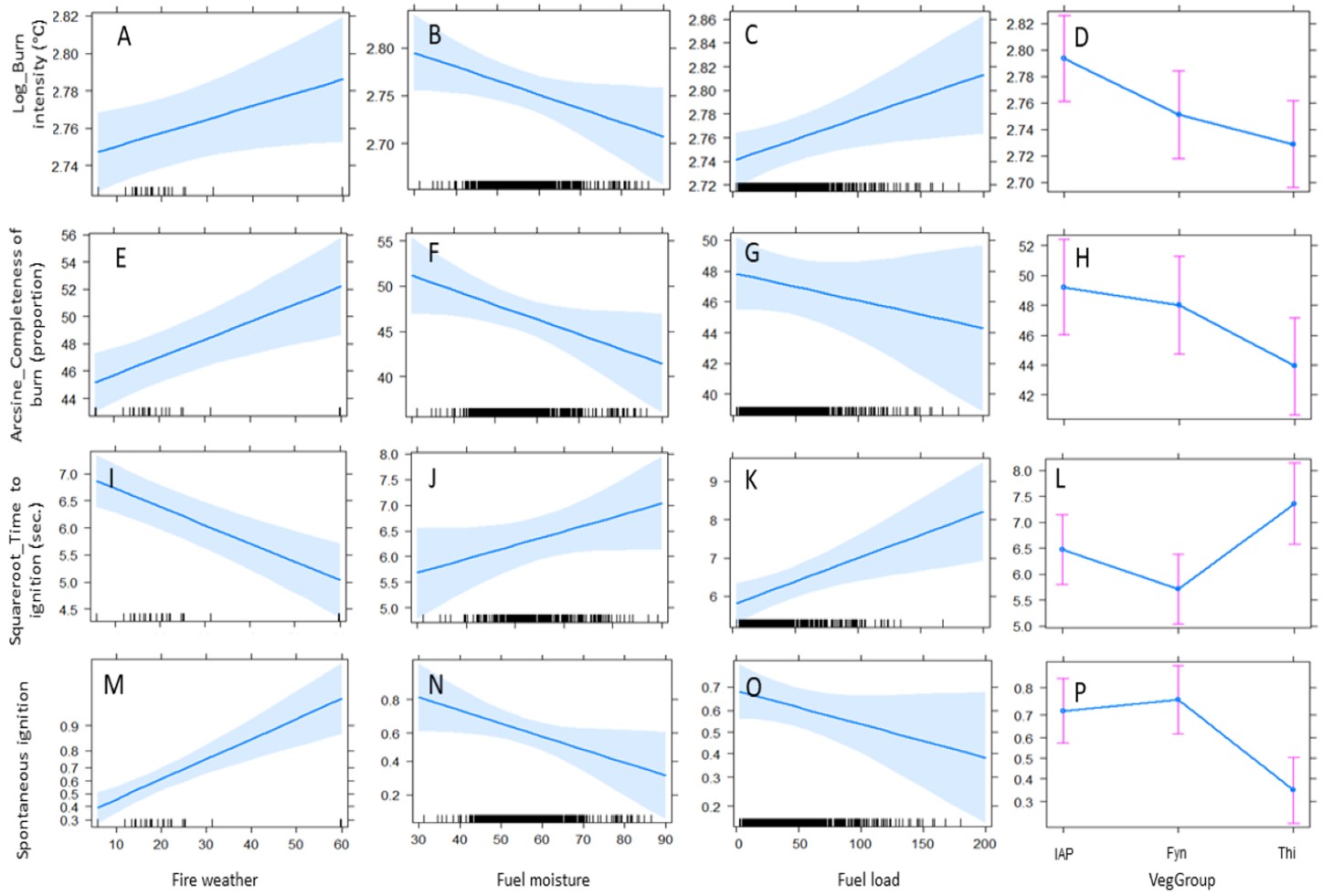

**Figure 1 Predicted effects of fixed factors on the flammability measures, (A–D) burn intensity, (E–H) completeness of burn, (I–L) time-to-ignition, and the probability of (M–P) spontaneous ignition.** Fixed factors were fire weather, fuel moisture, fuel load, and vegetation group (IAPs, invasive alien plants; Fyn, fynbos; and Thi, thicket). The effects shown here were based on the model outputs shown in Table 1 (shaded bands depict standard errors and whiskers show 95% confidence intervals).

(12–20%), except in terms of spontaneous ignition where vegetation groups and fire weather were most influential.

## Dried plant samples

Drying out of samples under ambient conditions for two weeks resulted in an average reduction in fuel moisture contents of approximately 30% (Fig. 2A), and the extent of this reduction did not differ significantly among vegetation groups ($H_2 = 1.4$, $p = 0.505$). Dried samples exhibited increased flammability compared to their live counterparts, that is, an average increase in burn intensity of 115 °C; an 11% increase in completeness of burn; and a 46 seconds reduction in time-to-ignition (Figs. 2B–2D). However, this differential response in flammability between dried and live samples was comparable among the vegetation groups in terms of burn intensity ($H_2 = 0.8$, $p = 0.666$), completeness of burn ($H_2 = 1.8$, $p = 0.410$), and time-to-ignition ($H_2 = 0.6$, $p = 0.741$).

**Table 1 Output of generalized linear mixed-effects models and logistic regression model that assessed flammability in terms of burn intensity, completeness of burn, time-to-ignition and spontaneous ignition.**

| Factors | Burn intensity | | | Completeness of burn | | | Time-to-ignition | | | Spontaneous ignition | | |
|---|---|---|---|---|---|---|---|---|---|---|---|---|
| | Estimate | Chisq[a] | Scaled estimate[b] | Estimate | Chisq[a] | Scaled estimate[b] | Estimate | Chisq[a] | Scaled estimate[b] | Estimate | Chisqa | Scaled estimate[b] |
| Fire weather | 0.0007 | 4.1* | 0.06731 | 0.1300 | 11.0*** | 0.1175 | −0.0339 | 21.0*** | 0.19471 | 0.0650 | 23.8*** | 0.6671 |
| Fuel moisture | −0.0015 | 4.4* | 0.11524 | −0.1616 | 4.6* | 0.1225 | 0.0271 | 2.8 | 0.09747 | −0.0379 | 4.5* | 0.3265 |
| Fuel load | 0.0004 | 5.6* | 0.10090 | −0.0180 | 1.1 | 0.0477 | 0.0121 | 9.3* | 0.16529 | −0.0063 | 2.6 | 0.1900 |
| Veg group (IAP and Fyn) | −0.0427 | 8.1* | 0.39141 | −1.1895 | 5.7 | 0.1048 | −0.7575 | 9.6** | 0.36388 | 0.2156 | 16.3*** | 0.2156 |
| Veg group (IAP and Thi) | −0.0648 | | 0.59446 | −5.2832 | | 0.4657 | 0.8838 | | 0.42458 | −1.5563 | | 1.5564 |
| Conditional $R^2$ [c] | 0.2961 | | | 0.2442 | | | 0.3983 | | | 0.3459 | | |
| Marginal $R^2$ [c] | 0.0942 | | | 0.0798 | | | 0.1935 | | | 0.2258 | | |
| $R^2$ (1\|Species)[c] | 0.2019 | | | 0.1644 | | | 0.2048 | | | 0.1201 | | |

Notes:
Fixed factors included in the generalized linear mixed-effects models (gaussian family, identity function; details in Table S2) and logistic regression model (binomial family, logit link function) were fire weather, fuel moisture, fuel load, and vegetation groups (IAPs, invasive alien plants; Fyn, fynbos; and Thi, thicket), while species was included as a random factor.
Significance codes: *$p < 0.05$, **$p < 0.01$, ***$p < 0.001$.
[a] Chisq statistics and significance levels were obtained from deviance tables (Type II Wald chi-square tests; details in Table S3).
[b] Scaled estimates were derived from incorporating the scale function in the generalized linear mixed-effects models and logistic regression model.
[c] $R^2$ values were derived using the $R^2$ GLMM function, where conditional $R^2$ indicates the proportion of variance explained by fixed and random factors combined, marginal $R^2$ indicates the proportion of variance explained by fixed factors alone and $R^2$ (1\|Species) indicates variance explained by the random factor alone.

## DISCUSSION

### Effects of fuel moisture, fire weather and fuel load on flammability

Fuel moisture content is widely regarded to be a major determinant of flammability in grassland, shrubland and forested ecosystems with sufficient evidence of its dampening effects on fire behaviour and flammability (*Bianchi & Defossé, 2015*; *Fares et al., 2017*; *Pausas & Paula, 2012*). That is why several fire danger indices attempt to account for the moisture contents of dead and live fuels to improve fire danger forecasting (*Chuvieco, Aguado & Dimitrakopoulos, 2004*; *Madula, 2013*; *Rothermel, 1983*; *Ruffault et al., 2018*; *Sirca et al., 2018*). Although dead fuel moisture responds closely and rapidly to fire weather, the relation between live fuel moisture and fire weather is more complicated as it depends on plant physiology and medium-to long-term meteorological trends (*Bianchi & Defossé, 2015*; *Bowman, French & Prior, 2014*; *Chuvieco, Aguado & Dimitrakopoulos, 2004*; *Nolan et al., 2016*). Accordingly, live fuel moisture in this study was not significantly correlated with fire weather in any of the study species. Live fuel moisture did significantly (negatively) correlate to burn intensity and completeness of burn, but the magnitude of its influence on flammability relative to the other factors investigated was generally low.

Fire weather significantly enhanced all measures of flammability, however the lack of response of live plant moisture contents to fire weather suggests that the mechanism through which fire weather enhances flammability may not be live fuel moisture. Other studies that have investigated fuel moisture–flammability relations (*Bianchi et al., 2018*) have not evidently assessed the effects of fire weather or have manipulated fuel moisture through drying out of fuels beyond natural levels of fluctuation in live fuels

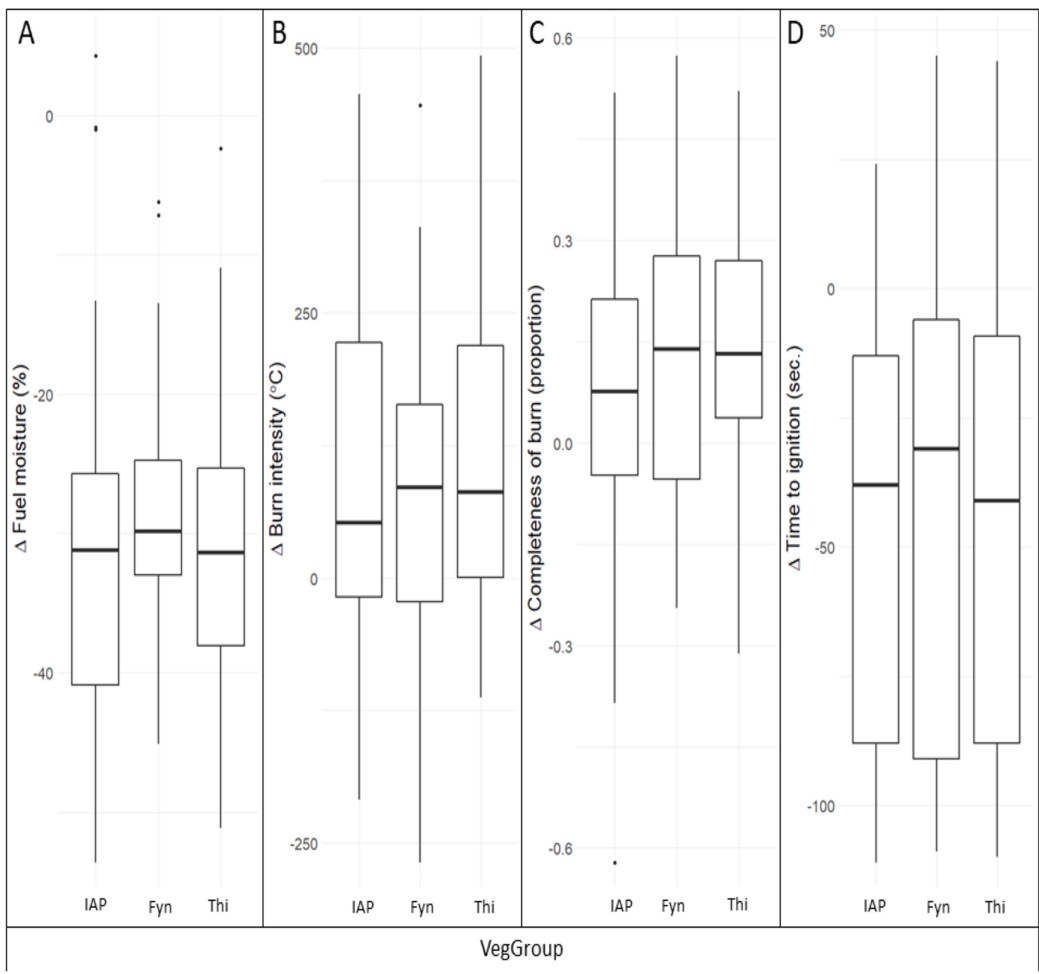

**Figure 2 The change (Δ) between live and dried samples in (A) fuel moisture, (B) burn intensity, (C) completeness of burn and (D) time-to-ignition, compared among vegetation groups.** Live and dried samples were of the same species under comparable fire weather conditions. Vegetation groups were IAPs, invasive alien plants; Fyn, fynbos; and Thi, thicket. Medians (lines), 25–75 quantile ranges (boxes), 1.5 * interquartile ranges (whiskers), and outliers (dots) are shown.

(*Dimitrakopoulos & Papaioannou, 2001*). We argue that the importance of live fuel moisture for flammability of evergreen shrublands rests on inter-specific and inter-vegetation type differences in fuel moisture contents (cf. *Chuvieco, Aguado & Dimitrakopoulos, 2004*), rather than medium-term intra-specific fluctuation in live fuel moisture in response to weather conditions. The incorporation of satellite-derived proxies for live fuel moisture into fire danger indices is therefore unlikely to be useful in these systems. Although fire weather increased all measures of flammability (and particularly ignitability), it was less influential than vegetation groups (see scaled estimates in Table 1). The contribution of short-term weather conditions to the severity of the 2017 Knysna fires was regarded to have been secondary to that of the long-term drought preceding these fires that would have caused a buildup of dead fuels (*Kraaij et al., 2018*). Fire weather is expected to increase in importance in its effects on flammability if

cognizance is taken of dry or dead fuels (see below) and when considering stand level fire behaviour. Although flammability experiments at the plant shoot scale are an improvement over those on excised leaves, and although results at the plant shoot and whole-plant scale are often in agreement (*Pausas & Moreira, 2012*), the scale of experimentation relative to stand or landscape level fire remains inadequate. For instance, particular aspects of fire weather, such as wind speed, greatly influence wildfire spread and spotting behavior (*Forsyth et al., 2019*). Such dynamics cannot be considered using the shoot-level flammability methods used in the current study; this may lead to an underestimation of the importance of fire weather on flammability and, by implication, fire behavior.

Fuel load had varying effects on flammability, depending on the measure considered; it increased burn intensity, but reduced ignitability. These findings support other evidence for positive correlations between the amount of biomass (~fuel load) that vegetation presents and fire intensity or severity (*Baeza et al., 2002*; *Keeley, 2009*; *Saura-Mas et al., 2010*), but negative correlations between fuel load and ignitability (*Guijarro et al., 2002*), the rate of spread (*Grootemaat et al., 2017*) and completeness of burn (*Kraaij et al., 2018*; *Van Wilgen, Higgins & Bellstedt, 1990*). Such contrasting effects on the different aspects of flammability relate to variation in fuel structural traits and emphasize the need to consider flammability in terms of its constituent measures rather than treating it as a composite measure (*Engber & Varner, 2012*; *Pausas et al., 2012*; *Santana & Marrs, 2014*).

Although live fuel moisture content, fire weather conditions, and fuel load had significant effects on some of the flammability measures, these factors did not explain a large portion of variability in the flammability response. Species, which was assessed as a random factor, often accounted for more variation in flammability than the fixed factors combined. This suggests important species effects on flammability, which warrant more detailed investigation. Our method of placing plant shoots horizontally on the barrel cavity grill with different amounts and sizes of plant parts oriented towards the grill could have introduced additional variation in the flammability response.

## Vegetation group effects in relation to fire risk

Vegetation group comparisons showed that the flammability of IAPs exceeded that of thicket in terms of all flammability measures and exceeded that of fynbos in terms of burn intensity. These findings support claims (*Forsyth et al., 2019*; *Stander, 2019*) and other evidence (*Brooks et al., 2004*; *Kraaij et al., 2018*; *Richardson & Rejmánek, 2011*) that invasions by alien plants can add to the severity, intensity, and difficulty of control of wildfires. Fynbos and IAPs were more ignitable than thicket, and thus present higher risks under moderate and high fire weather conditions, whereas thicket presents lower risks under low and moderate fire weather conditions. Accordingly, observations from the 2017 Knysna fires indicated that thicket only becomes ignitable under very high or extreme fire weather conditions but may then burn at intensities exceeding that in fynbos but not that of IAPs (*Kraaij et al., 2018*) presumably on account of disparate fuel loads (*Keeley, 2009*; *Mandle et al., 2011*). In our study, there were no significant differences between the flammability of fynbos and IAPs but completeness of burn appeared to be the highest in

fynbos. *Kraaij et al. (2018)* also observed that fynbos burnt more completely than thicket and IAPs in the 2017 Knysna fires which suggests that the risk of recurring fire will be lowest in fynbos for some period post-fire, whereas incomplete burning of IAPs will not afford the same level of risk reduction shortly post-fire.

### Simulated drought conditions

Extremely large and severe fires, including the 2017 Knysna fires, are often associated with preceding droughts (*Kraaij et al., 2018*; *Quinn, 1994*; *San-Miguel-Ayanz, Moreno & Camia, 2013*; *Williams, 2013*) and the resultant increase in dead fuels (*Keeley, 2009*). The extent and severity to which thicket, normally regarded as a fire-resistant (~poorly ignitable) vegetation (*Calitz, Potts & Cowling, 2015*; *Cowling & Potts, 2015*), burnt in the 2017 Knysna fires was attributed to extreme fire weather conditions and to the preceding severe drought (*Kraaij et al., 2018*). In this study, we confirmed that the drying of fuels as a crude proxy for severe drought effects considerably increased flammability. However, the magnitude of the increase in flammability in response to drying of fuels was consistent across vegetation groups. Flammability, and by implication fire risk, is thus unlikely to increase disproportionately in one vegetation group compared to another under extended drought unless the production of dead fuels due to drought would be disproportionate among the vegetation groups. We concede that the proxy for drought conditions could not realistically simulate all potential effects of drought on fuel modification and flammability, and in particular on the dying off of fuels and resultant increase in litter component. Detailed consideration of this aspect was beyond the scope of this study and warrants further investigation. Given the low moisture contents of dead fuels, the ratio of dead to live fuels are likely to be a useful indicator of fire risk in evergreen shrublands (*Keeley, 2009*). Proxies for this ratio should, therefore, be sought for incorporation into fire danger indices.

## CONCLUSIONS

Our experimental burning of shoots of 30 shrub species confirmed that fire weather, live fuel moisture, and fuel load have significant effects on flammability measures. However, vegetation group and species differences accounted for most of the variation in flammability. Flammability was generally highest in invasive alien plants, intermediate in fynbos, and lowest in thicket. The drying of plant shoots resulted in increases in flammability that were comparable among vegetation groups, implying that under drought conditions, fire risk should not increase disproportionately in one vegetation group compared to another, unless the production of dead fuels is disproportionate among vegetation groups.

## ACKNOWLEDGEMENTS

We would like to extend gratitude to Phillip Frost for assisting with the Canadian Fire Weather Index calculations, Tatenda Mapeto for making weather data available, and to Ian Ritchie for constructing the flammability apparatus. Tiaan Strydom, Nicole Blignaut, Natasja Van Zyl, Marita Burger, Corne Brink, Siyabonga Sibisi, Gert Botha, Jooste Eileen

and Herman Viviers assisted with specimen collection and flammability experiments. Constructive comments of two anonymous reviewers led to improvements to the manuscript.

### Funding
Funding for this study was provided by the African Centre for Coastal Palaeoscience, Postgraduate research scholarship from the Nelson Mandela University, and German Academic Exchange Service—South African National Research Foundation Masters scholarship. The funders had no role in study design, data collection and analysis, decision to publish, or preparation of the manuscript.

### Grant Disclosures
The following grant information was disclosed by the authors:
African Centre for Coastal Palaeoscience, Postgraduate research scholarship from the Nelson Mandela University.
German Academic Exchange Service—South African National Research Foundation Masters scholarship.

### Competing Interests
Alastair J. Potts and Tineke Kraaij are Academic Editors for PeerJ.

### Author Contributions
- Samukelisiwe T. Msweli performed the experiments, analyzed the data, prepared figures and/or tables, authored or reviewed drafts of the paper, and approved the final draft.
- Alastair J. Potts conceived and designed the experiments, analyzed the data, authored or reviewed drafts of the paper, and approved the final draft.
- Herve Fritz analyzed the data, authored or reviewed drafts of the paper, and approved the final draft.
- Tineke Kraaij conceived and designed the experiments, analyzed the data, authored or reviewed drafts of the paper, and approved the final draft.

### Data Availability
The raw data are available in the Supplemental Files.

### Supplemental Information
Supplemental information for this article can be found online at http://dx.doi.org/10.7717/peerj.10161#supplemental-information.

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
