# Peer review of "Fire weather effects on flammability of indigenous and invasive alien plants in coastal fynbos and thicket shrublands (Cape Floristic Region)"

_PeerJ, doi:10.7717/peerj.10161_

## Round 0.1 · original submission · Major Revisions

This manuscript has now been reviewed by two experts in the field and the outcome was mixed. Reviewer 1 expresses fundamental concerns and skepticism about the validity of findings from burn experiments, such as were conducted in this manuscript. Reviewer 2 sees value in the study, but also points out a number of areas in which the manuscript should be improved. I note that this manuscript cites a publication (entitled "Plant flammability experiments offer limited insight into
vegetation-fire dynamics interactions") that discusses limitations of burn experiments. I welcome the authors to revise and resubmit their manuscript, provided that they address all of the concerns from both reviewers.

Reviewer 1 ·

Basic reporting

No comment, the manuscript complies with the standards.

Experimental design

Methods require a few clarifications. There are some relevant issues:
- Moisture content variation after sampling and its relation with weather;
- The FWI as a suitable indicator of weather influence on flammability;
- Fire intensity as a temperature;
- Fuel samples for moisture content determination exposed to a temperature lower than desirable;
- Fuel moisture content expressed wrongly.
Details are provided in the general comments section.

Validity of the findings

The findings suffer from the problems identified above. More generally, flammability experiments are known to be poor surrogates of "real" fire conditions. Any finding from an experiment of this type is always subject to strong doubts about its validity.

Additional comments

Flammability experiments are a difficult topic, because they tend to be subjective and the researchers involved tend to have poor understanding of fire processes. I did identify a number of issues in the manuscript:
L59-63. Some confusion here between weather and climate. Individual fires are never climate-driven, the fire regime is.
L75-76. This sentence does not make sense. The only reason why RH and temperature are used in fire danger rating is because they determine fuel moisture content.
L118. Live fuel moisture? Because fire weather determines dead fuel moisture.
L153-154. "subsequently spreading fire to other plant structures" is not really accurate nor relevant. Better to just say that fine fuels such as plant shoots are the primary carriers/vectors of fire spread.
L159. What defined a sample? One stem?
L164. The experiments deal with live fuels. Live fuels lose moisture rapidly after collection. So, either you ensured the samples maintained their moisture status or you let the moisture vary after collection. In one case or another the moisture content is not expected to correlate with the current fire weather at the time of the burning. Were the experiments carried out outdoors?
L166. A Canadian reference should be used, e.g. Van Wagner (1987). As the study deals with live fuels the authors should have used other indices of the Canadian system which should be better correlated with live fuel moisture content, namely the Drought Code (DC) or the Buildup Index (BUI), as the FWI integrates this plus the atmospheric influences. Better yet, the authors could have tried to separate the effect of the atmosphere (by using the vapour pressure deficit, currently used as a predictor of live fuel moisture content) and the effect of drought.
L187. Although unfortunately found quite often in the fire ecology literature, this is extremely wrong: maximum temperature does not reflect fire intensity or the heat flux produced by combustion, as all flames attain the same temperature. Temperatures measured in fires are not the temperature of the fire, which is meaningless by definition, but the temperature of the measuring device. Using the measurements of the study, fire intensity would be expressed by the product of time-to-ignition (assuming it is an acceptable proxy for fire spread rate) and the proportion of burnt fuel, i.e. following the fire intensity concept of Byram (1959).
L200. Current knowledge indicates that drying temperatures lower than 100ºC (Matthews in the Int. J. Wildland Fire) underestimate fuel moisture content.
L201-202. Do you mean that it was calculated on a wet weight basis? The standard is to refer fuel moisture content always on a dry basis ((wet-dry)*dry*100).
L203. Make the sentence more accurate: fire intensity is directly proportional to fuel load. Keeley (2009) is not the right ref. here and you should cite Byram (1959).
L205. So you had % fuel consumed and fuel load, which enables calculation of fuel consumption (in actual weight) from which fire intensity is calculated.
L203. Reference to running GLMM appears here a 2nd time.
L293. Also in response to drought (water availability).
L301-302. Very unlikely! There are other motives but note that the fuel samples were pre-heated and this would totally override any effect that temperature could have had.
L311-312. Flammability experiments are microscale experiments that do not warrant this type of inference.
L317-323. Basically you are undermining the results here by acknowledging that they are representative of real world conditions.
L350-354. This is totally unwarranted. Burn completeness as an outcome of microscale experiments cannot be compared with that resulting from a wildfire. Under wildfire conditions the amount of residual fuel is mostly a function of fuel size (particle diameter or thickness) and condition (moisture).
L373. A more relevant motive is that dead fuel moisture content is much lower under conditions that allow fire spread.

·

Basic reporting

No comment

Experimental design

All good. There should be mention in the discussion of how accurate the method was.

Validity of the findings

There is some confusion about the aims and the conclusions.

Additional comments

This study examines the flammability of three different vegetation types in the Cape region of South Africa and how flammability responds to fire weather. It is an important contribution and in general the methods are appropriate, it is well written with appropriate references. However, I think some of the logic was loose, which left me confused about what the actual hypothesis and conclusions were. In particular, there seems to be no distinction made between live fuel moisture and dead fuel moisture.

Specific points.

Line 60. I think ‘fires’ should be ‘fire regimes’. An individual fire is weather-driven, not climate driven.

Line 60-63. I think this could be clearer. For example, Bradstock 2010 explains in arid areas, dryness limits fuel and fires follow episodic rain, whereas in temperate forests, fuels are not limiting, and fire follows drying of those fuels. Bradstock, R (2010) A biogeographic model of fire regimes in Australia: current and future implications. Global Ecology and Biogeography 19, 145-158.

Line 66. Replace ‘would’ with ‘may’. This is only a theory.

Line 87. Add ‘and’ before ‘changes’.

Line 97. I think here you need to add something about litter fuels. Fires are often an interaction between litter and live fuels. For example, in eucalypt forests, it is generally thought that you cannot have a canopy fire without a corresponding surface fire. If this is the case in your system, then you are not measuring everything you need to about the flammability of these vegetation types. If litter fuels are not important, then say so.

Line 111. Add ‘it’ after ‘showed’.

Line 117. These aims are not very clear. Surely comparing vegetation types is one of the main aims. Why pose those hypotheses, when broadly speaking, they have been established in previous studies?

L146. What form are these invasive species? Presumably small trees, but still taller than the natives? Do they co-occur with the natives, or replace them?

L160. It is important to describe how soon the burning occurred after sampling. E.g. if they were all sampled at once and then burned one by one during the day, then some would have dried out more than others.

Line 204. How was this dry mass calculated? Using the ratios found in the subsequent study of dried plant samples?

Line 287. This is not true. Dead fuel moisture (e.g. in leaf litter) is responsive to weather. Live fuel moisture responds to drought (as your 2-week drying shows). Those fire danger indices account for dead fuel moisture. See for example Nolan et al 2016. Here is a quote from there: “The first major novel outcome of this study was the formal demonstration that dynamic transformations in fuel moisture associated with major wildfires can occur rapidly for dead fuels and within several weeks to months for live fuels”. Nolan, RH, Boer, MM, de Dios, VR, Caccamo, G, Bradstock, RA (2016) Large-scale, dynamic transformations in fuel moisture drive wildfire activity across southeastern Australia. Geophysical Research Letters 43, 4229-4238.

Line 293. Similarly, you would not expect live fuel moisture to be related to fire weather.

Line 308. I don’t think anyone argues the latter.

Line 310. I disagree. They are very useful for tracking fire risk through drought, and as you explain later, the 2017 fires were partially caused by drought. They are not a component of hourly or daily fire weather, but they are a component of weekly or monthly flammability fluctuations.

Line 324. The reason that fuel load is negatively related to ignitability is probably to do with leaf traits like thickness and surface area to volume ratio. This has been studied in some detail. This is not my area of research so I don’t know the studies, but I suggest you look one or two up.

Line 333. It strikes me that your method can be a source of variation and you should acknowledge that. For example, each time you put a branch on the apparatus, a different amount is touching the plate. Can you show this variation, e.g. in graphs of the actual data?

Line 372. I like that you are considering this, but it is complicated. Most (but not all) dead leaves will drop to become litter, and litter has different flammability properties than leaves on the tree (probably a lot less flammable due to packing).

Figures. It would be nice to have a study area figure, either a map, or photos of the vegetation types or both.

---

## Round 0.2 · accepted · Accept

Thank you for your thorough job in addressing the concerns of the reviewers.

Reviewer 1 ·

Basic reporting

No problems.

Experimental design

Ok.

Validity of the findings

The authors improved the manuscript, although the basic conceptual problems persist. As usual in this subject of study, the paper will likely please flammability people and will be ignored by fire behaviour people.

·

Basic reporting

The paper is clear, the methods appropriate and the use of literature is good.

Experimental design

The minor issues I raised in the first review have been addressed

Validity of the findings

The minor issues I raised in the first review have been addressed

Additional comments

The authors have done a thorough job of responding to the reviews.